# The isocyanide $S_N2$ reaction

Pravin Patil [1,2,4], Qiang Zheng[2,4], Katarzyna Kurpiewska[3] & Alexander Dömling [1,2] ✉

The $S_N2$ nucleophilic substitution reaction is a vital organic transformation used for drug and natural product synthesis. Nucleophiles like cyanide, oxygen, nitrogen, sulfur, or phosphorous replace halogens or sulfonyl esters, forming new bonds. Isocyanides exhibit unique C-centered lone pair σ and π* orbitals, enabling diverse radical and multicomponent reactions. Despite this, their nucleophilic potential in $S_N2$ reactions remains unexplored. We have uncovered that isocyanides act as versatile nucleophiles in $S_N2$ reactions with alkyl halides. This yields highly substituted secondary amides through in situ nitrilium ion hydrolysis introducing an alternative bond break compared to classical amide synthesis. This novel 3-component process accommodates various isocyanide and electrophile structures, functional groups, scalability, late-stage drug modifications, and complex compound synthesis. This reaction greatly expands chemical diversity, nearly doubling the classical amid coupling's chemical space. Notably, the isocyanide nucleophile presents an unconventional Umpolung amide carbanion synthon (R-NHC(-) = O), an alternative to classical amide couplings.

The amide bond is one of the most common functional and structural elements, as the backbones of all peptides and proteins and almost every second drug are composed of amide bonds. Thus, the construction of amide bonds is fundamental to organic synthesis as it provides access to the backbone of pharmaceuticals, agrochemicals, natural products, peptides and proteins and functional materials (Fig. 1a). Recent data mining charting the amine–acid cross-coupling space revealed that there are numerous opportunities for reaction discovery[1]. Nonetheless, the great majority of amide groups are formed by the coupling of a nucleophilic amine and an electrophilic carboxylic acid building block. Notably, the direct coupling of amines and carboxylic acid is unfavorable, hence requires an activated ester which is formed by aggressive, expensive or waste-full reagents (Fig. 1b)[2]. Far less common ways to form amides involve, for example, oxidative and radical-based methods or the alkylation of nitriles to yield nitrilium ions with in situ hydrolysis[3,4]. Based on our longstanding interest in isocyanide chemistry, we asked ourselves to what degree the known boundaries of isocyanide nucleophilicity can be pushed to exploit new and synthetically useful reactivities. The isocyanide has

both a σ-type *C*-centered HOMO and a *C*-centered π-type LUMO which accounts for the unusual reactivity of isocyanides (Fig. 1c)[5]. An example of this is its ability to act both as a nucleophile and an electrophile in the α-addition of a nucleophile and an electrophile onto the same functional group atom, the isocyanide-*C*, which is a quite unusual feature in organic chemistry, and accounts for many reactions of isocyanides such as multicomponent reactions and heterocycle syntheses[6–8]. Due to their unusual *C*-only centered reactivity, isocyanides were also coined stereochemical chameleons[9].

Isocyanides in multicomponent reaction chemistry (IMCR) is probably the most famous application. It involves the nucleophilic attack to the electrophilic oxo and imine-*C* and the subsequent addition of an internal or external nucleophile onto the nitrilium-*C*, followed by a rearrangement and is well established in the Passerini and Ugi reactions[7,10]. Motivated by the powerful IMCR, which is based on the unusual reactivity of the isocyanide-*C*, we designed a novel multicomponent reaction (Fig. 1d). Our design involves an initial nucleophilic attack of the isocyanide on an alkyl halide in the sense of a nucleophilic substitution reaction ($S_N2$); the intermediate nitrilium ion

[1]Institute of Molecular and Translational Medicine, Faculty of Medicine and Dentistry and Czech Advanced Technology and Research Institute, Palacký University in Olomouc, Olomouc, Czech Republic. [2]Department of Drug Design, University of Groningen, Groningen, The Netherlands. [3]Department of Crystal Chemistry and Crystal Physics Faculty of Chemistry, Jagiellonian University, 30-387 Kraków, Poland. [4]These authors contributed equally: Pravin Patil, Qiang Zheng. ✉e-mail: alexander.domling@upol.cz

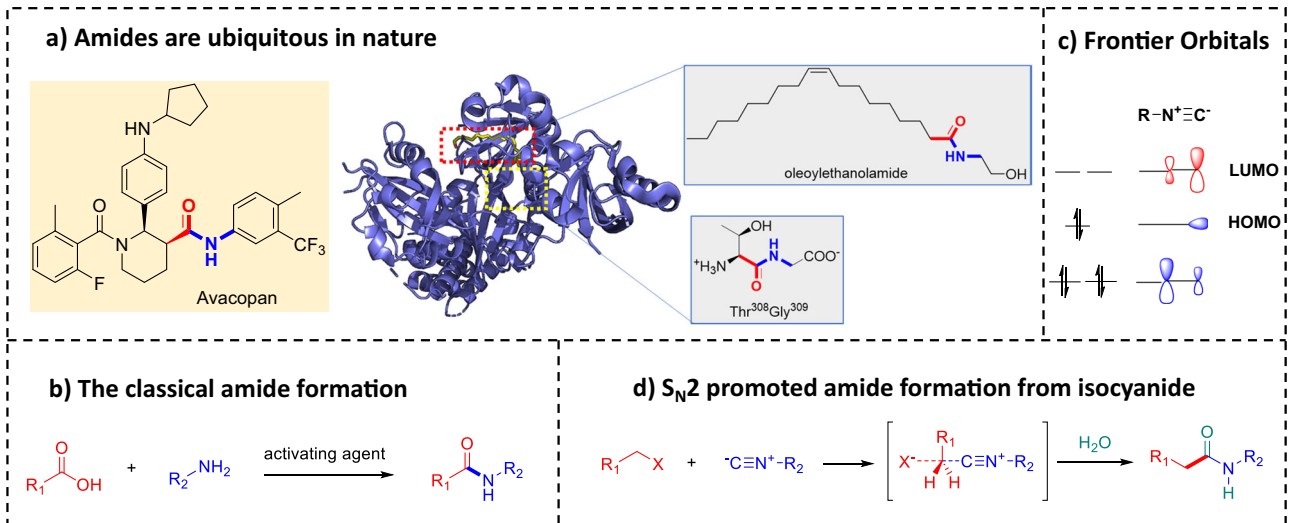

**Fig. 1 | Background and motivation for novel amide formation. a** Amides are ubiquitous in nature: autoimmune disease drug Avacopan and transcription factor hypoxia-inducible factors HIF-3α (PDB-ID 7V7W) with natural ligand oleoyletha-nolamide. **b** The classical approach to amide formation involves stoichiometric activation/coupling of carboxylic acid and amine. **c** Frontier orbitals of the iso-cyanide to rationalize chameleonic behavior as a *C*-nucleophile and *C*-electrophile. **d** The novel multicomponent reaction amide synthesis involves an $S_N2$-promoted *C-C* coupling reaction, followed by hydrolysis of the nitrilium ion.

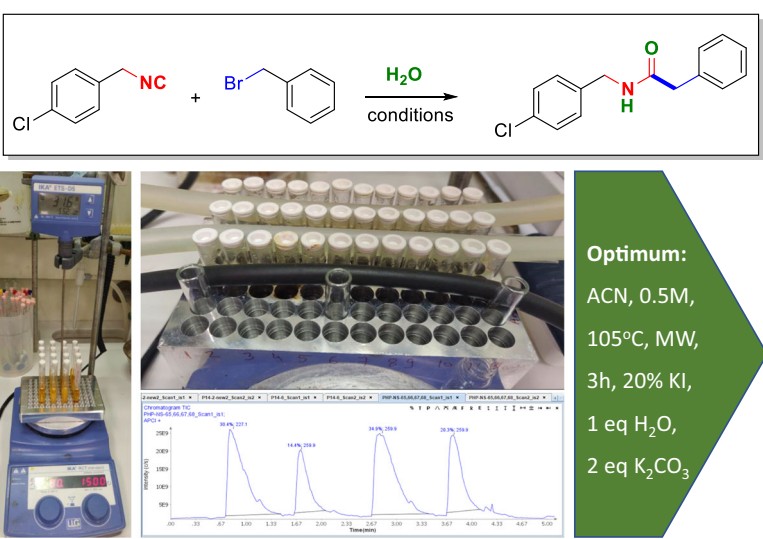

**Fig. 2 | Optimization of reaction parameter using high-throughput experimentation.** Scheme of the model reaction, parallel heating in a metal block, and stagged HPLC injections.

then reacts with water to give the stable amide species. Conceptually and experimentally, the reaction design is not obvious, and potential issues preventing the planned reaction outcome could involve i.a. insufficient isocyanide nucleophilicity, premature hydrolysis of the isocyanide or the alkyl halide.

## Results

### Reaction optimizations

The $S_N2$ reaction is known to be very sensitive to the substrate struc-tures, as well as reaction conditions[11–13]. While initial attempts to run the new reaction were promising by mass spectrometry analysis, the yields were far from being synthetically useful. Taking into account the established condition knowledge of the $S_N2$ reaction, we first interrogated stoichiometry and ratio of the reactants, temperature, temperature source, and solvents employing high-throughput

experimentation (HTE)[14,15]. HTE methods used were parallel reactions in 96, 48, and 24-well format, parallel heating in a metal block, parallel TLC analytics, and stacked injection into SFC. The methods are described in more detail in the Supplementary Information. We chose the model reaction of *p*-chloro benzyl isocyanide with benzyl bromide, a good electrophile in $S_N2$ reactions and good visibility of educts and product in TLC (Fig. 2). Next, we investigated the result of additives in the $S_N2$ reactions. Biphasic phase transfer catalysts (PTC) were often used in $S_N2$ reactions to increase yields and conversion[16]. We screened 16 different common PTCs (Supplementary Information). The addition of iodine salts is often described as advantageous in the $S_N2$ reactions as it converts the less reactive chloride-leaving groups into the more reactive iodo-leaving group. After thorough optimization of all para-meters, the optimized conditions involved the microwave heating at 105 °C for 3 h of 1:2 ratio of isocyanide, alkyl halide, 20 mol% KI

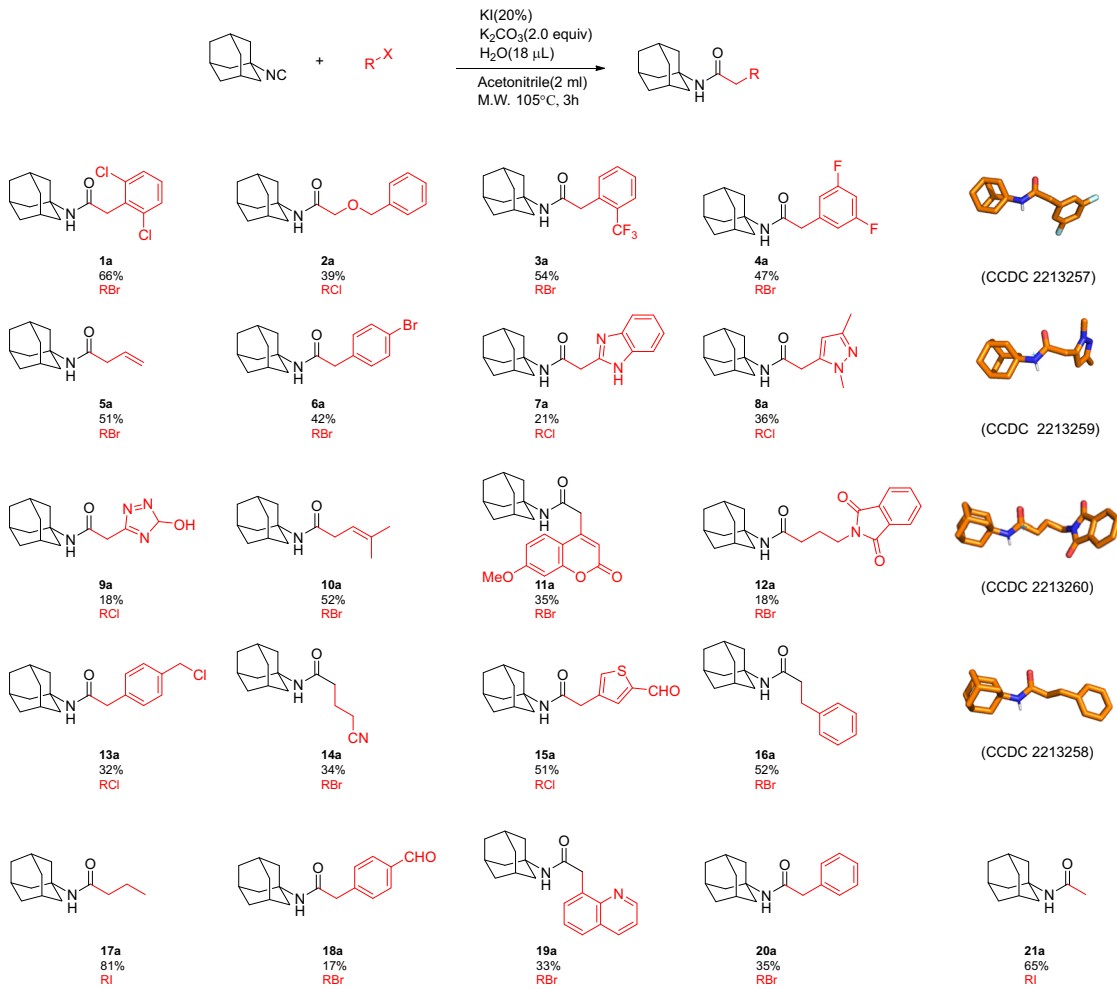

**Fig. 3 | Scope of the halide electrophile with adamantyl isocyanide as the fixed component.** Several stick presentations of X-ray structures and their CCDC codes are given.

catalyst, and 1 equivalent of water in acetonitrile in the presence of 2 equivalent of the inorganic base $K_2CO_3$ (Fig. 2).

## Scope and limitations

The substrate scope for this reaction is very broad (Figs. 3–5). With the optimized conditions in hand, we interrogated the scope of the halide with respect to the leaving group, sterical bulkiness, electronic nature and building block diversity. Among the halide leaving group, chloride, bromide and iodide reacted well according to the well-established leaving group trend I>Br>Cl. To test the functional group tolerance, we successfully reacted 21 different alkyl halides with adamantyl isocyanide on a mmol scale (Fig. 3). Adamantyl isocyanide is a solid, non-smelling, bench-stable powder that has been synthesized recently on a mol scale[17]. A variety of alkyl halides with different functionalities were well tolerated. The small methyl group can be easily introduced (**21a**), whereas bulky alkyl groups or alkyl groups with β-branching do not react. Long-chain alkyl groups can be introduced (**17a**), also with a terminal phthalic amide amine protecting group (**12a**), whereas Boc-protecting groups were found to be not stable under microwave conditions (Supplementary Information). For several alkylation products, single crystals revealed X-ray structures that support the structural identity (**4a, 8a, 12a, 16a**). Allyl (**5a**) and benzyl (**1a, 3a, 4a, 6a, 13a**) groups react well due to the conjugated nature of the pentagonal bipyramidal transition state, as suggested by the classical $S_N2$ literature. Specifically to mention, is bis benzyl chloride derived **13a**, which can be mono alkylated in 32% yield and can be potentially

further reacted through the unreacted benzyl chloride. Also, the nature of the heterocyclic structures that could be reacted is quite diverse, including benzimidazole (**7a**), pyrazole (**8a**), triazole (**9a**), phthalimide (**12a**), coumarin (**11a**), thiophene (**15a**), and quinoline (**19a**). Especially to mention are **15a** and **18a**, which are formed from bifunctional (hetero)aromatic benzyl chloride benzaldehydes. The aldehyde functionality can be further derivatized, as will be shown below. We also found substrates that did not give the expected products or gave very low yields (<30%, Supplementary Information).

The evaluation of the isocyanides also revealed a broad scope (Fig. 4). We reacted 20 different isocyanides with methyl iodide in satisfactory to good yields. Benzylic (**23a, 24a, 25a, 29a, 30a**), aromatic (**31a, 33a, 34a, 35a, 36a, 37a**), aliphatic (**27a, 42a**) and heteroaromatic (**26a, 28a, 32a**) isocyanides all worked well. When isocyanides with a basic side chain were reacted, we observed the double alkylation and a quaternary amine salt formation (**38a, 39a**). Noteworthy, also α-amino acid isocyanides (**40a, 41a**) worked well.

We performed a number of mixed examples to further elaborate the scope and usefulness of the reaction (Fig. 5). Highly substituted **47a** is especially noteworthy, as it comprises a combination of a sterically hindered α,α-disubstituted cyclopropyl benzyl isocyanide with a bifunctional 4-formylbenzyl chloride. The new method is also applicable to the facile synthesis of diverse lipid derivatives (**56a, 60a, 61a**), which could be of interest in lipidomics applications. Bulky isocyanides (**47a, 50a, 54a**) and phenyl isocyanides with bulky o-substitutents (**43a, 51a, 52a, 53a, 55a**) reacted nicely. Amide

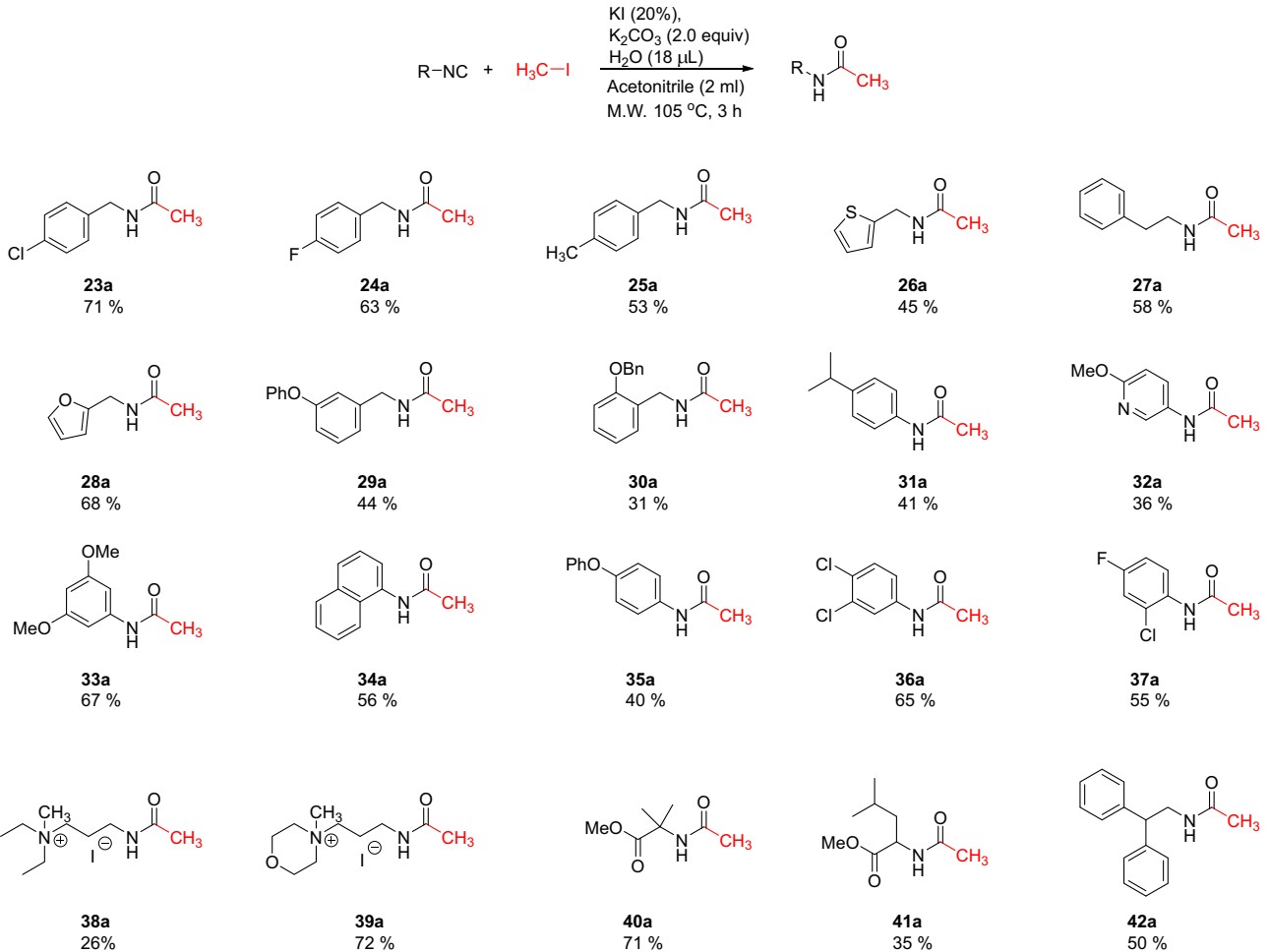

**Fig. 4 | Scope of isocyanide nucleophile with methyl iodide as the fixed component.** The figure encapsulates the versatility of this reaction by showcasing the successful reaction of 20 distinct isocyanides with methyl iodide, yielding satisfactory to good yields.

**55a** is accessible with a free compatible benzylic hydroxyl group. 4-Methylpentenoic acid (pyroterebic acid) ester or amides are common in biologically active isoprenoid compounds from plants. Compound **54a** is a pyroterebic acid amide, and it comprises an unprecedented synthesis. Another example of the incorporation of an isoprenoid side chain (homo geranyl acid) is exemplified in **60a**. It is conceivable that this methodology can be used to incorporate isotope labeled carboxy-*C* via the isocyanide. In summary, complex structures can be accessed from simple available building blocks in one step.

### Scaling and late-stage functionalization

To further stress the reaction performance, we evaluated the robustness of this reaction toward pharmaceutical late-stage diversification on an actual drug[18]. Late-stage-functionalization is a drug discovery technique to selectively derivatize already complex 'drug-like' molecules and is used to further improve their properties[18]. Phenoxybenzamine (dibenzyline) is an alpha blocker used for the treatment of hypertension. To establish the usefulness of our novel S_N2 reaction, we reacted dibenzyline with adamantyl isocyanide and were able to isolate the expected amide product **22a** in 40% yield (Fig. 6).

A gram-scale application of the isocyanide to amide transformation further underscores the robustness of the reaction. We performed the synthesis of thiophene carbaldehyde with fair yields. By reacting 1.59 g of bifunctional 5-(chloromethyl)thiophene-2-carbaldehyde with 1.61 g of adamantyl isocyanide on a 10 mmol scale (Fig. 6), we obtained product **15a** in a 51% yield (1.57 g). The aldehyde group can be further functionalized to create complex molecules with just a few steps. For

this purpose, we used multicomponent reactions (MCR) for further derivatization of the thiophene carbaldehyde[7,10,19]. Compound **15a** is of interest to test further reactivity due to its unprotected aldehyde group based on the functional group compatibility of the reaction. Thus, we used **15a**, each in a Ugi-4CR, a Groebke Blackburn Bienaymé (GBB-3CR) reaction, and a Ugi tetrazole reaction to exemplify the rapid increase of molecular complexity (Fig. 6). The Ugi-4CR product **1b** was obtained in 72% yield in one step from easily available building blocks. Noteworthy, an alkynyl amide is introduced in a straightforward mild manner. Electrophilic alkynyl amides are often used in covalent drug discovery targeting cysteines, and an alkynylamide substructure can be found in the FDA-approved Acalabrutinib Bruton's tyrosine kinase targeting drug[20]. Next, we investigated aldehyde **15a** as a substrate in the GBB-3CR reaction. The GBB-3CR is a popular method to synthesize highly substituted bicyclic imidazo heterocycles, which already have proven their value as drugs and candidates[21]. Thus, we reacted 2-aminopyridine with aldehyde **15a** and cyclohexyl isocyanide in a GBB-3CR under microwave conditions in methanol to obtain complex heterocycle **1c** in 36% yield. Lastly, we performed a Ugi tetrazol reaction employing aldehyde **15a**. Tetrazoles are often used as advantageous carboxylic acid bioisosteres and can be broadly obtained by multicomponent reaction chemistry[19].

In summary, the new S_N2 reaction turned out to be scalable, useful in late-stage-functionalization and can yield highly interesting intermediates for allowing further chemistries to increase structural diversity in a quasi-exponential complexity increase, in just three steps: isocyanide synthesis, S_N2 reaction, further aldehyde reaction.

**Fig. 5 | Mixed reaction examples of the isocyanide nucleophile with alkyl halide.** The figure presents a diverse array of mixed reaction examples showcasing the dynamic interplay between the isocyanide nucleophile and the alkyl halide. These illustrative instances collectively increase the versatility and application potential of the reaction.

## Mechanism and chemical space

Our preliminary observations support an $S_N2$-type mechanism. Accordingly, the nucleophile isocyanide attacks from the backside to form a trigonal bipyramidal transitions state **I** and kicks out the leaving halogen anion. The intermediately formed nitrilium ion **II** undergoes water attack on the isocyanide-C **III** and, through tautomerization, reveals the final amide **IV** upon hydrolysis.

Several lines of evidence, including kinetic analysis (Supplementary Information), support an $S_N2$ mechanism: sterically hindered substrates such as neopentyl iodide or isobutylbromide do not give any reaction product; the reaction is strongly solvent dependent and runs well in the polar solvent DMF which are believed to stabilize the transitions state, but not in apolar toluene or protic methanol; the reaction rate depends on the nature of the nucleofuge as reported in the $S_N2$ literature I>Br>Cl (Supplementary Information). To exclude a possible radical mechanism, we performed the reaction in the presence of 2x stoichiometric amounts of the radical quencher TEMPO and did not find any difference in the reactivity (Supplementary Information)[22,23]. While running the reaction in the absence of water and direct injection in the mass spectrometer, we could observe a strong peak corresponding to the bromo nitrilium ion (see below). Another hallmark of the $S_N2$ mechanism is the stereochemical inversion of the electrophilic carbon center called Walden inversion. To investigate the stereochemical inversion, we conducted experiments on chiral halides (Fig. 7). Unfortunately, the (S)-2-bromobutyric acid as the halide source did not yield the desired results, probably due to the incompatibility of the free carboxylic acid. Thus, we prepared the

corresponding methyl ester. To prevent any further racemization of the ester, we evaporated all the solvents and utilized the crude methyl ester of (S)-2-bromobutyric acid for the subsequent $S_N2$ reaction with adamanty isocyanide under standard conditions. The resulting product, **62**, was isolated in 22% yield. To assess the enantiomeric ratios, we employed chiral supercritical fluid chromatography mass spectrometry (SFC-MS) on the product. We found that two enantiomers of **62** were formed in a 27:73 ratio, which is in accordance with an $S_N2$ mechanism (Supplementary Information). While one would expect full inversion in an $S_N2$ reaction, the ~2:1 enantiomeric ratio could be explained twofold; the product is racemization prone due to the two CH flanking carbonyl substituents and the basic reaction conditions, or the reaction is running according to a mixed mechanism involving direct inversion of the bromide substrate or intermediate formation of the iodide substrate and therefor double inversion (retention). The exact nature of the mechanism needs additional future investigation.

In conclusion, there is strong evidence that the reactions run according to an $S_N2$ mechanism (Fig. 8a). In the classical amide coupling approach, the carbonyl is a carbocation synthon, while in the $S_N2$ approach the rare amide carbanion synthon is the result of an Umpolung (Fig. 8d). The isocyanide is commonly synthesized from its primary amine precursor (Ugi method: formylation > dehydration or Hoffman reaction)[10,24]. Alternatively, isocyanide can be obtained by reductive amidation with formamide (Leukart-Wallach) and dehydration from an aldehyde or ketone precursor (Fig. 8c)[25,26]. This $S_N2$ reaction involves coupling a primary amine with a C1 synthon derived from chloroform (Hoffmann) or formic acid (Ugi) with an alkyl halide

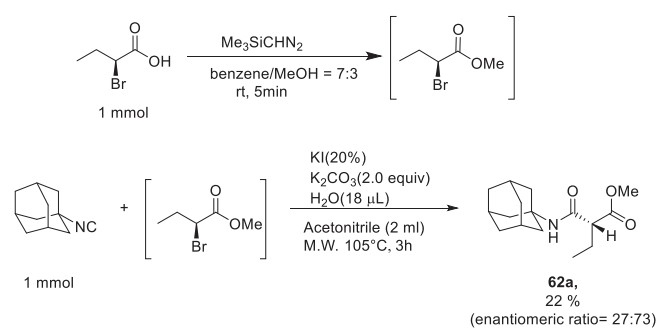

**Fig. 6 | Late-stage diversification, scale-up and some follow-up chemistries.** Robustness was validated through pharmaceutical late-stage diversification on phenoxybenzamine (dibenzyline), yielding amide product **22a**. Scale-up demonstrated isocyanide-to-amide transformation yielding product in Ugi-4CR, GBB-3CR, and Ugi tetrazole reactions, showcasing rapid complexity increase.

**Fig. 7 | S$_N$2 Mechanism investigation for stereochemical inversion.** Stereoselective reaction of methyl ester of (S)-2-bromobutyricacid methyl ester with adamantly isocyanide under standard S$_N$2 reaction condition.

or coupling an aldehyde/ketone with an NC synthon derived from formamide (Leukart-Wallach) with an alkyl halide (Fig. 8c). The wide availability of primary amines, aldehydes, ketones, and alkyl halides makes this reaction highly valuable for synthesis. Notably, unlike classical S$_N$2 reactions that use simple nucleophiles (e.g., halides, CN-, thio- or alcoholates), the described S$_N$2 reaction utilizes the structural diversity and complexity of isocyanides (Fig. 5). This is leading to a strong increase in structural complexity upon coupling with alkyl halides (e.g., **60a**). Next, we asked the question whether the new reaction can access a chemical space different from the classical amide coupling. For this, we investigated the commercial availability of the corresponding carboxylic acid needed to form the target amides and compared them with the corresponding halide (Supplementary Information). Surprisingly, in 52% of cases, the corresponding

carboxylic acids were not commercially available at all. Noteworthy, in the remaining 48%, the carboxylic acid was, on average, 2.4 times more expensive than the corresponding chloride. It turned out that the chemical space accessible by the two orthogonal amide syntheses is very different, and only 12% are overlapping (i.e., can be synthesized by both methods). In conclusion, our herein-reported novel S$_N$2 reaction is of high synthetic value as it allows access to a chemical space that otherwise can only accessed through time-consuming and lengthy multistep syntheses and leads to a strong increase in molecular complexity, otherwise uncommon in S$_N$2 reactions.

## Discussion

Arguably, the amide bond formation is among the most important reactions in organic chemistry. The value of the amide group in organic chemistry cannot be overstated. It is on top of the most frequent functional groups occurring in bioactive molecules described in medicinal chemistry literature[27]. More than 1/2 of the marketed drugs contain at least one amide group. Thus, the amide bond formation is the most practiced reaction in medicinal chemistry and one of the most frequently used in process chemistry[2]. While the classical amide coupling is a powerful reaction, there are many more hypothetical ways in which amides can be formed, with each new transformation imprinting a unique accessibility fingerprint on the product. The discovery of novel reactivities is key to leverage untapped chemical space and to broaden the toolbox used in medicinal and other chemistries[28], exemplified by a recently described copper-catalyzed deaminative esterification with a broad scope[29]. The use of isocyanides in S$_N$2 reactions is such an example of unprecedented reactivity giving access to unusual, otherwise difficult-to-synthesize amides. Classically the amide group is constructed from a carboxylic acid derivative and a primary or secondary amine using specific activation conditions, and a

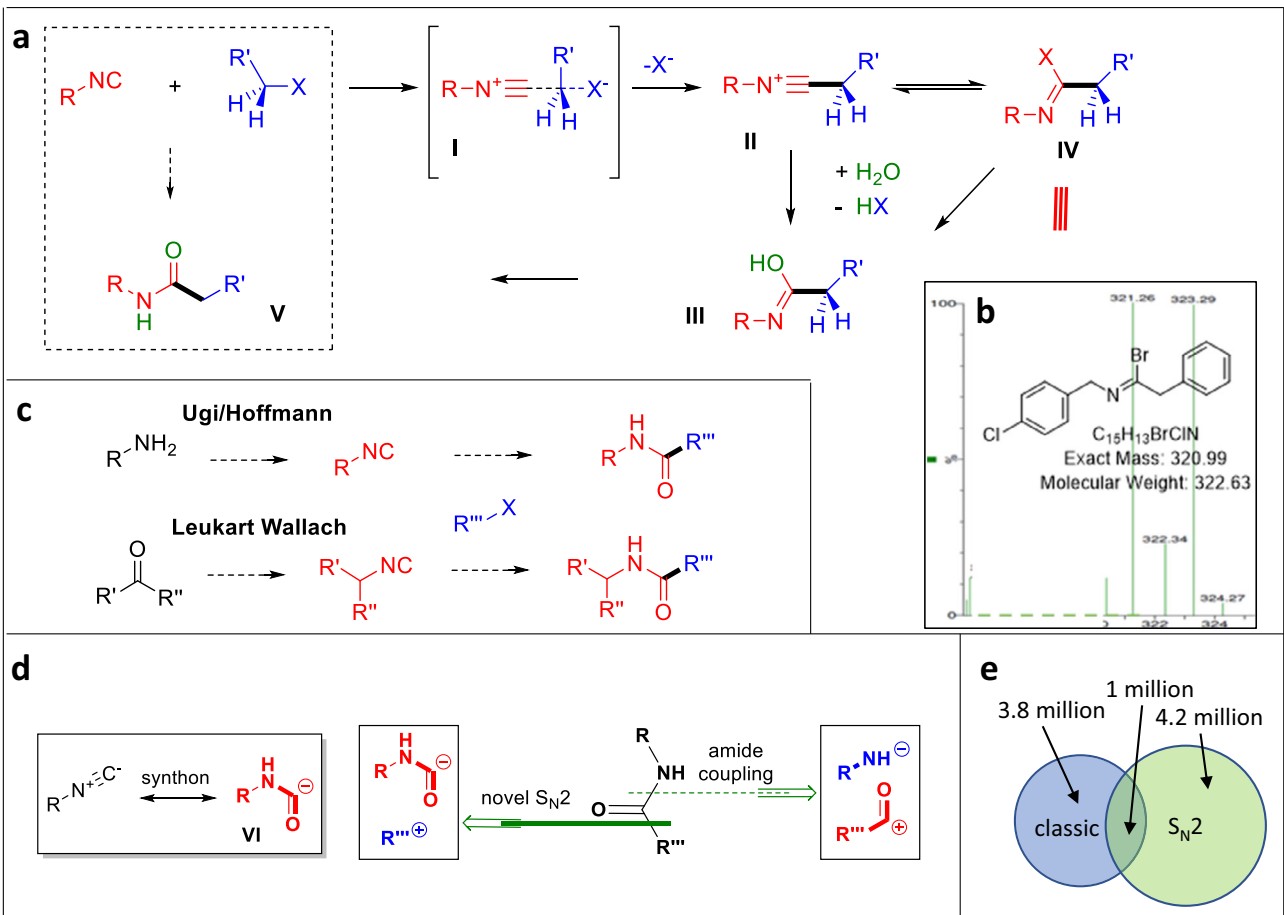

**Fig. 8 | Proposed mechanism and evidence, RNC synthon, retrosynthetic valuation, and comparison with the classical amide coupling. a** Proposed mechanism and MS evidence of an imidoyl bromide intermediate (**b**). **c** Most popular and efficient access to isocyanides by two different synthesis pathways. **d** Comparison of the classical and the $S_N2$-based amide formation and the amide carbanion isocyanide synthon. **e** Venn diagram for chemical space analysis of the classical and $S_N2$-based amide formation and its overlap.

plethora of aggressive, expensive, and waste-full coupling reagents have been described[2]. Therefore, sustainable and alternative amidations have emerged as an important synthetic strategy to exploit the commercial and natural prevalence of the amide functional group[30,31], leading us to consider the transformation of an isocyanide into an amide by alkylation and hydrolysis. The reaction was specifically designed to complement the popular amide coupling reaction.

Here, we show for the first time that isocyanides can be alkylated by an $S_N2$ mechanism through a nitrilium ion with concomitant hydrolysis to the corresponding amide. In this novel 3-component reaction, the isocyanide can be described as a Umpolung-derived rare amide carbanion synthon. The use of isocyanides as acyl anion equivalents provides a conceptually innovative approach to amide synthesis. As isocyanides are most commonly derived from either primary amines or aldehydes or ketones, the new reaction connects a primary amine via a formyl-C to an alkyl halide or, in the second case, an aldehyde or ketone carbonyl is connected through the formamide-C to an alkyl halide. The position of the amide group in the classical amide coupling of amines and carboxylic acids and the herein-described isocyanide/alkylation-derived amides are different. By repurposing halide building blocks as amides instead of relying on the traditional amide coupling method, a subtle but significant transformation occurs in terms of synthetic accessibility. Attempting to synthesize the same molecules by the two orthogonal methods, large-scale data analysis of educts reveals a great disadvantage of the classical method since the required carboxylic acid building blocks are

more difficult to access, not available at all and more expensive than the corresponding alkyl halides by an average factor of 2.4 (Supplementary Information). We also performed a survey of commercial availability of the required building blocks from a commercial vendor catalog (Fig. 8e, Supplementary Information), which revealed that 3.8 million amides are accessible by the classical method and 4.2 million by the new method, with only 1 million matched molecular pairs between the two sets. A chemoinformatic analysis of commercial building blocks demonstrates that by utilizing halides and primary amine-, aldehyde-, or ketone-derived isocyanides, our method more than doubles the available amidation chemical space. There is minimal overlap of chemical space compared to the classical amide coupling, demonstrating that a halide-isocyanide amidation can provide broad access to new and complementary structures. Repurposing of halide and isocyanide building blocks provides an enormous opportunity to expand the accessible chemical space or amides because halide and amine or aldehyde/ketone feedstocks are typically low cost and available in high diversity. A halide-isocyanide amidation would therefore leverage the abundance of one popular building block and easily access isocyanides from other popular building blocks. Collectively, these analyses quantify the value that a halide-isocyanide amidation would provide as an addition to the synthetic toolbox. High-throughput experimentation was used to develop the reaction, along with classic scope studies, both of which demonstrated robust performance against many pairs of reactants. The new reaction can be carried out under practical, mild conditions with yields ranging from

good to moderate to poor, depending on the structure of the reactants. The functional group compatibility of the reaction is high. Late-stage functionalization of a drug is exercised. Alkyl halides are very frequently encountered in pharmaceutical research, so harnessing this functional group would also provide plenty of opportunities for late-stage diversification. Upscaling of the reaction to gram scale has been shown. Complex, otherwise difficult-to-access compound classes such as lipids, isoprenoids or functionalized amino acids can be synthesized by this method. Electrophilic alkyl halides are a cheap, abundant feedstock and are commercially available in high diversity, making them a valuable starting material for amide synthesis. Similarly, isocyanides are accessible in two steps from abundant primary amines or ketones or aldehydes. In conclusion, we have successfully developed an efficient new way to form amides by reacting isocyanides, alkyl halides, and water in a three-component fashion. Our innovative amide synthesis will be of potential use in the synthesis and discovery of novel bioactive molecules. It is beyond current amide bond forming methods, and with its unusual synthon features, exponential complexity increase, and its wide scope, it will allow us to scout novel chemical space hitherto inaccessible[32]. It is conceivable that other nucleophiles than water can also react with the intermediate-formed nitrilium ion in this new multicomponent reaction, and work is currently underway in our laboratory to broaden the range of nucleophiles.

## Methods

### General procedure for compound 1a-61a

To a microwave vial equipped with a magnetic stir bar containing isocyanide (1.0 mmol), alkyl methyl halide (2.0 mmol), potassium iodide (0.2 mmol), potassium carbonate (2.0 mmol), $H_2O$ (1.0 mmol) and acetonitrile (0.5 M) were added. The reaction vessel was sealed and irradiated in the cavity of the microwave reactor at the set temperature of 105 °C for 3 h. Upon completion of the reaction, the vial was cooled to room temperature. The reaction mixture was diluted with dichloromethane 5 ml and filtered off to remove the inorganic solids. Solvents were evaporated under vacuum to give a crude reaction product. The crude product was purified by flash chromatography on silica gel to obtain the pure product.

### Synthesis of compound 1b

The mixture of aldehyde (15a, 1.0 mmol), tert-butyl isocyanide (1.0 mmol), aniline (1.0 mmol) and but-2-ynoic acid (1.0 mmol) in 1.0 ml of methanol was stirring under room temperature for 12 h. The crude product was purified by flash chromatography on silica gel (petroleum ether/ethyl acetate, 3:1) to get product 1b (392 mg, 72%).

### Synthesis of compound 1c

To a microwave vial equipped with a magnetic stir bar containing pyridine-2-amine (1.0 mmol), cyclohexyl isocyanide (1.0 mmol), aldehyde (15a, 1.0 mmol), methanol (1.0 ml) and scandium triflate (0.1 mmol) were added. The reaction vessel was sealed and irradiated in the cavity of the microwave reactor at the set temperature of 100 °C for 2 h. Upon completion of the reaction, the vial was cooled with a stream of air. The resulting reaction mixture was diluted with ethyl acetate (50 ml), thoroughly washed with 1 M HCl (2 × 50 ml) and saturated $Na_2CO_3$ (50 ml) aqueous solutions, dried over $Na_2SO_4$ and concentrated under reduced pressure. The crude product was dissolved in ethyl acetate (5 ml) and subjected to flash chromatography with silica (25 g) and ethyl acetate/pentane (3:7) as eluent to give pure 1c (175 mg, 36%).

### Synthesis of compound 1d

The mixture of aldehyde (15a, 1.0 mmol), isocyanide (1.0 mmol), aniline (1.0 mmol) and TMSN₃ (1.0 mmol) in 1.0 ml of methanol was stirred under room temperature for 12 h. The crude product was purified by chromatography on silica gel (petroleum ether/ethyl acetate, 3:1) to get product 1d (221 mg, 39%).

### General procedure for the control experiment ruling out a radical mechanism

To a microwave vial equipped with a magnetic stir bar containing adamantyl isocyanide (1.0 mmol), 2-(bromomethyl)-1,3-dichlorobenzene (1.0 mmol), potassium iodide (0.2 mmol), potassium carbonate (2.0 mmol), TEMPO (2.0 mmol), $H_2O$ (1.0 mmol), and acetonitrile (2.0 ml) were added. The reaction vessel was sealed and irradiated in the cavity of the microwave reactor at the set temperature of 105 °C for 5 h. Upon completion of the reaction, the vial was cooled at room temperature. The crude product was purified by chromatography on silica gel (petroleum ether/ ethyl acetate, 3:1) to get product 1a (198 mg, 59%).

## Data availability

The experimental method and data generated in this study are provided in the Supplementary Information file. The crystallographic data for structures of 4a, 8a, 12a and 16a have been deposited in the Cambridge Crystallographic Data Centre under accession CCDC codes 2213257, 2213259, 2213260 and 2213258, respectively. Copies of the data can be obtained free of charge via www.ccdc.cam.ac.uk/data_request/cif. All other data are available from the authors upon request.

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

## Acknowledgements

This research has been supported (to A.D.) by ERA Chair Accelerator Grant agreement ID: 101087318. Q.Z. is supported by a Chinese Scholarship Council PhD fellowship. Robin van der Straat performed the chiral SCF experiments.

## Author contributions

A.D. conceived the research project and raised funding. P.P. and Q.Z. conducted the experiments. P.P., Q.Z., and A.D. analyzed the data. A.D. conceptualized and directed the project and drafted the manuscript with assistance from all coauthors. All authors contributed to part of the experiments and/or discussions. K.K. determined the single crystal X-ray structure.

## Competing interests

The authors declare no competing interests.
