## [Peer Review File · Nature Communications]

The Isocyanide SN2 ReactionReviewers' Comments:

Reviewer #1:

Remarks to the Author:

In this manuscript P. Patil et al describe the discovery and optimization of the novel isocyanide S_N2 reaction to form amide bonds. They describe the motivation for a novel amide formation reaction and outline the high throughput experimentation to optimize the reaction conditions. These are described in detail. They synthesized ~60 different amides to describe the scope and limitations of the novel isocyanide amide formation reaction. In addition, they synthesized larger quantities of amides and applied late-stage-functionalization to proof the utility of the reaction. A reaction mechanism is proposed. A cheminformatics approach was used to enumerate the chemical space of the new reaction and compare it with the traditional active ester reaction of amide formation.

The manuscript is well written and concise. It is scientifically convincing and a important finding, especially for medicinal chemistry.

I'm missing a more detailed analysis of the reaction mechanism, especially the quantum mechanical calculation of transition states and associated free energies. Provided these are amended the manuscript should be published in Nature Communications.

Reviewer #2:

Remarks to the Author:

Domling and co-authors describe an intriguing conversion of isocyanides to amides via an S_N2 displacement followed by hydrolysis of the intermediate nitrilium by trapping with water. The reaction is intriguing because of the simplicity of preparing amides from an alkyl halide or sulfonate through the attack by an isocyanide.

Although the reaction has not been previously reported there are several reports of similar processes. For example, photoredox coupling of bromoacetates with isocyanides affords the same nitrilium intermediate that is hydrolyzed to the corresponding amide (J. Org. Chem. 2020, 85, 14077–14086). What is unique in the current manuscript is the direct displacement without the need for catalysis. The question then becomes, "how effective is the method?" The authors have performed an intensive study of the reaction scope (Figures 3-5) using high-throughput experiments. In Figure 3 adamantylisocyanide is used as the prototype nucleophile under optimized conditions to afford amides in yields ranging from 17-81% with an average of 42% yield. Figure 4 varies the isocyanide in a displacement with methyl iodide, a particularly reactive electrophile; the yields range from 26% to 72% with an average yield of 52%. Examples in Figure 5 combine a variety of isocyanides with alkyl halides for which most are activated (benzylic or allylic halides); the yields range from 8-64% with an average yield of 35%. Figures 3-5 demonstrate that a variety of functional groups tolerate the reaction; the example of the amide synthesis on scale presented in Figure 6 is impressive.

The challenge with the reaction scope is that the yields are modest compared to conventional methods of preparing amides. The intensive attention devoted to the synthesis of amides means that numerous, high-yielding methods are available making a high bar for new methods. The authors counter the attractiveness of their method by arguing that the precursors are orthogonal to those typically used to prepare amides, acids and amines. While true, there are relatively few commercially available isocyanides for the authors' S_N2 displacement method though many halides are available as the other component. On balance, most chemists are unlikely to embrace the S_N2 -isocyanide displacement method in preference to existing methods of preparing amides because of the modest yield and the limitation requiring primarily alkyl halides, preferably activated, for the S_N2 displacement.

If the reaction assembly is only modestly efficient, then how novel is the mechanism? The mechanism is surprising in being so simple and working for isocyanides derived from tertiary amines. For example, Figure 3 employs adamantyl isocyanide which might cleave to the carbocation after the S_N2 displacement. Perhaps the unaccounted mass for the reactions in Figure 3 comes from trapping with water, though no information was provided as to the remaining mass. Several benzylic isocyanides

similarly afford amides (Figures 4 and 5) without fragmenting to benzylic cations, the mass balance notwithstanding.

Lastly, the presentation of the research. The manuscript is too long with information that is of limited relevance (see for example lines 154-159, 200-209, and much of the discussion section). A shorter, more focused presentation would have been more compelling.

On balance, the research is very interesting. However, the modest yields, limited scope, and wordy presentation detract from the discovery. Given where related amide-forming reactions have been published, I am ambivalent about advocating for publication in Nat. Commun.

Reviewer #3:

Remarks to the Author:

This manuscript by Dr. Patil et al described "The Isocyanide SN2 Reaction". They discovered the weak nucleophilicity of the lone pair of electrons of the isocyanic carbon, which reacts with halides and water to form an SN2 reaction of amides through the high throughput experimentation screening method, and optimized the reaction conditions. The substrate expansion of the reaction and the post-modification of complex molecules to form amides were demonstrated using the optimized conditions in mild to good yield ; Meanwhile, the evidences for a related SN2 reaction mechanism is also partially provided. This is an important discovery in isocyanic chemistry and an effective complement to the method of synthesizing amides.

However, I have three major concerns to raise:

- 1) As the authors mentioned, there are many chiral compounds containing amide bonds in bioactive substances, therefore, due to the concern that the reaction conditions in the paper are relatively harsh, could the authors do a reaction to synthesize a peptide bond and observe whether its stereostructure is maintained?
- 2) In terms of mechanistic proof, could the authors use secondary halides as substrates (especially the carbon substituted by halogens are chiral carbons) to observe their Walden inversion phenomenon?
- 3) It is recommended to adjust some expressions in the discussion part, otherwise the experimental evidence cannot fully support the conclusion. For example, the authors mentioned "A chemoinformatic analysis of commercial building blocks demonstrates....." in page 11, If the authors have not done a verification experiment of chirality retention when the chiral compound forms an amide bond, it needs to be cautiously expressed.

In addition, there are some minor errors that need to be revised:

- 1) In page 1, line 29, "the classic amid coupling"?
- 2) In page 4, line 40, "by coupling of the"?
- 3) In page 9, line 209, "of isocyanides (Fig.4)", is it Fig.5?
- 4) In the supplementary part, S3, method development part, "We run he model reaction"?
- 5) In the supplementary part, S21, 4a, "3.43 (s, 1H)"?
- 6) In the supplementary part, some spectra are not complete and clear. For example, S40, ¹HNMR of compound 3a, S41, ¹HNMR of compound 4a, et al. Please check all.
- 7) In the supplementary part, related solid compounds lack melting point data. In short, there are

many writing and editing errors in this edition of the manuscript and SI, and the above may only be part of them, and the authors are expected to revise it carefully.

This is an interesting work, however, the manuscript can only be considered for publication in Nature Communications after the points above being addressed thoroughly and an overall re-evaluation thereafter.

REVIEWER COMMENTS

Reviewer #1 (Remarks to the Author):

In this manuscript P. Patil et al describe the discovery and optimization of the novel isocyanide S_N2 reaction to form amide bonds. They describe the motivation for a novel amide formation reaction and outline the high throughput experimentation to optimize the reaction conditions. These are described in detail. They synthesized ~60 different amides to describe the scope and limitations of the novel isocyanide amide formation reaction. In addition, they synthesized larger quantities of amides and applied late-stage-functionalization to proof the utility of the reaction. A reaction mechanism is proposed. A cheminformatics approach was used to enumerate the chemical space of the new reaction and compare it with the traditional active ester reaction of amide formation.

The manuscript is well written and concise. It is scientifically convincing and a important finding, especially for medicinal chemistry.

I'm missing a more detailed analysis of the reaction mechanism, especially the quantum mechanical calculation of transition states and associated free energies. Provided these are amended the manuscript should be published in Nature Communications.

Answer: Thank you for the nice comments on the overall manuscript. We provide a good number of experimental proof for the S_N2 reaction, in form of reaction rate with various concentrations and time intervals, nitrilium ion mass spectrometry observation from reaction mixture, etc. We also provide proof of the Walden stereochemistry inversion for the S_N2 reaction in our manuscript. We believe that the quantum mechanical calculation will be a vast new research topic in this field of isocyanide S_N2 reaction of isocyanide and alkyl halide and is out of the scope of this already experimentally very rich manuscript. Moreover, the authors don't feel to be specialist in QM calculations. We will certainly approach capable specialists in the future to perform in depth QM calculations on the reaction mechanism.

Reviewer #2 (Remarks to the Author):

Domling and co-authors describe an intriguing conversion of isocyanides to amides via an S_N2 displacement followed by hydrolysis of the intermediate nitrilium by trapping with water. The reaction is intriguing because of the simplicity of preparing amides from an alkyl halide or sulfonate through the attack by an isocyanide.

Although the reaction has not been previously reported there are several reports of similar processes. For example, photoredox coupling of bromoacetates with isocyanides affords the same nitrilium intermediate that is hydrolyzed to the corresponding amide (J. Org. Chem. 2020, 85, 14077–14086).

Answer: We are pleased to hear the positive words of reviewer 2 about our work. However, we disagree with the claimed similarity of our method with the previously published visible-light photocatalytic addition of isocyanides onto N-methylene anilines with subsequent hydrolysis (J. Org. Chem. 2020, 85, 14077–14086). The reactions differ in all aspects including reaction mechanism, catalysis, starting materials, products, scope – they only have in common the use of the isocyanide as a reagent in novel reactions.

Our reaction is not using any metal catalyst, nor light, nor is it based on radicals, as we could experimentally show by addition of a radical scavenger. We also tested our method with phenacyl bromide as well as phenacyl chloride but we didn't observe the corresponding amide formation, instead of we isolate the hydrolyzed product of isocyanide and halides and unreacted isocyanide. Moreover, the reviewer is referring to the reference, J. Org. Chem. 2020, 85, 14077–14086, in which simple benzyl bromide, as well as benzyl isocyanides, isocyanide obtained from glycine esters failed to give any products, while in our method these substrates all give good yields. This shows that both methods are totally unrelated to each other. As it is shown in (J. Org. Chem. 2020, 85, 14077–14086) a radical reaction mechanism is followed, while our reaction is following the SN2 reaction pathway.

What is unique in the current manuscript is the direct displacement without the need for catalysis. The question then becomes, "how effective is the method?" The authors have performed an intensive study of the reaction scope (Figures 3-5) using high-throughput experiments. In Figure 3 adamantylisocyanide is used as the prototype nucleophile under optimized conditions to afford amides in yields ranging from 17-81% with an average of 42% yield. Figure 4 varies the isocyanide in a displacement with methyl iodide, a particularly reactive electrophile; the yields range from 26% to 72% with an average yield of 52%. Examples in Figure 5 combine a variety of isocyanides with alkyl halides for which most are activated (benzylic or allylic halides); the yields range from 8-64% with an average yield of 35%. Figures 3-5 demonstrate that a variety of functional groups tolerate the reaction; the example of the amide synthesis on scale presented in Figure 6 is impressive.

The challenge with the reaction scope is that the yields are modest compared to conventional methods of preparing amides. The intensive attention devoted to the synthesis of amides that numerous, high-yielding methods are available making a high bar for new methods. The authors counter the attractiveness of their method by arguing that the precursors are orthogonal to those typically used to prepare amides, acids and amines. While true, there are relatively few commercially available isocyanides for the authors' SN2 displacement method though many halides are available as the other component. On balance, most chemists are unlikely to embrace the SN2-isocyanide displacement method in

preference to existing methods of preparing amides because of the modest yield and the limitation requiring primarily alkyl halides, preferably activated, for the SN2 displacement.

Answer: There is no 'one-size-fits-all' method for the synthesis of amides. While amide synthesis is highly optimized even to an automated degree in peptide synthesizers; while many amide bonds can be formed in high yields employing standard methods; we would like to argue that many other amide formations outside the usual coupling 'comfort zone' are problematic and give low yields under standard conditions. Such examples are discussed in the critical review of Bradley et al. (Chem. Soc. Rev., 2009, 38, 606–631). Amide couplings that perform poorly with established protocols are relatively frequently encountered, in particular with sterically hindered substrates and electron deficient amines, and alternative strategies for synthesis of such amides have to be applied. The real numbers of unsuccessful amid bond formations under standard conditions remains unknown because of the habit of many chemists to only report only a handful of 'surprisingly' successful transformations and 'round up' isolated yields. Likely this 'dark amide' space is quite large. We could also have ameliorated the tables in our manuscript by purposely removing the low yielding examples and concentrate on methyl, ethyl, fusil, like practiced in so many publications. Immediately the average yields would skyrocket. But we are rather advocates of a 'fair' assessment of a reaction including a lot of functionalized, more complex substrates, beyond methyl, ethyl, propyl... We also reported purposely examples underscoring the SN2 mechanism. Here we are not surprised that the reaction shows a great variation of yields because it is well established that the classical SN2 reaction is highly dependent on sterical and electronic factors as well as solvent, catalyst etc. A major stumbling block of the introduction into daily practice of a literature described new reactions is the missing scope and limitations. As was pointed out exemplarily by Krska (Process Research & Development, Merck Sharp & Dohme Corporation), 'The structural complexity of pharmaceuticals presents a significant challenge to modern catalysis. Many published methods that work well on simple substrates often fail when attempts are made to apply them to complex drug intermediates.', or by Cernak et al., ' and 'Many synthetic methods, especially modern catalytic transformations, were developed using very simple model substrates and do not perform well on complex pharmaceutical intermediates. If not addressed, this disconnect between the types of substrates amenable to many catalytic methods (generally low-molecular-weight, lipophilic hydrocarbons with few functional groups) and the ideal substrates for drug synthesis (complex N-heterocycles with multiple hydrogen-bond-donor and -acceptor functional groups) can create a synthetic bias in medicinal chemistry programs toward making molecules with less drug-like properties, ultimately leading to lower-quality clinical candidates.'. For this reasons we believe it is of uttermost importance to show reasonable scope of a reaction by introducing diverse substrates and not too much focus primarily on yields. Yield improvement need a careful one-by-one optimization of the conditions which

can not be reflected in a generalized reaction condition, which is always a compromise to tackle many different reactions.

'The intensive attention devoted to the synthesis of amides that numerous, high-yielding methods are available making a high bar for new methods.'

If the classical amide coupling would be already perfect why the field would desperately investigate alternative amide couplings? Noteworthy, the ACS Green Chemistry Institute together with the ACS Petroleum Research Fund, both organizations highly funded by chemical industry, are supporting projects aiming to find alternative amide coupling methods. Especially from a large-scale industrial view, classical amide coupling suffers tremendously from sustainability issues, such as super stoichiometric use of expensive, hazardous coupling reagents resulting in wasteful procedures. For example, the synthesis of typical commercial peptides generates 34 tons of waste and 118 tons CO₂ eq. per kg of API. We are not claiming that our method is solving all current amide coupling problems, but we propose a novel innovative approach which has its pros and cons over other methods.

'The authors counter the attractiveness of their method by arguing that the precursors are orthogonal to those typically used to prepare amides, acids and amines. While true, there are relatively few commercially available isocyanides for the authors' S_N2 displacement method though many halides are available as the other component.' While isocyanides currently are not often used and their commercial availability is limited, in fact they can be very simply accessed in large numbers as shown by many researchers. Also, we showed recently highly convenient procedures for their parallel synthesis in 96-well plates (Isocyanide2.0, Green Chem., 2020,22, 6902-6911). In the same publication we also exemplified the mol-scale isocyanide synthesis in less than 1h from its corresponding formamides. Thus, it's possible to synthesis thousands of isocyanides this way as we do in my lab. In analogy, boronic acids before the successful introduction of Pd based couplings such as Suzuki-Miyaura where also extremely rare species in the vendors catalogues.

*'On balance, most chemists are unlikely to embrace the S_N2-isocyanide displacement method in preference to existing methods of preparing amides because of the modest yield and the limitation requiring primarily alkyl halides, **preferably activated**, for the S_N2 displacement.'* We disagree on the description 'preferably activated'. In fact, most of our substrates are not activated (allyl or benzyl).

If the reaction assembly is only modestly efficient, then how novel is the mechanism? The mechanism is surprising in being so simple and working for isocyanides derived from tertiary amines. For example, Figure 3 employs adamantyl isocyanide which might cleave to the carbocation after the S_N2 displacement. Perhaps the unaccounted mass for the reactions in Figure 3 comes from trapping with water, though no information was provided

as to the remaining mass. Several benzylic isocyanides similarly afford amides (Figures 4 and 5) without fragmenting to benzylic cations, the mass balance notwithstanding.

Answer: We mention several times in our manuscript the potential and observed side reactions such as isocyanide and halide hydrolysis. In fact, during the optimization of the reactions in an earlier phase when using water in excess we mostly observed hydrolysis, irrespective of the employed isocyanide and halide. Notably, we never observed the carbocation formation of adamantyl group and never observed any adamantyl alcohol when we used adamantyl isocyanide; similarly, neither any other benzyl or other isocyanides led to carbocation-mediated hydrolysis to the alcohols.

Lastly, the presentation of the research. The manuscript is too long with information that is of limited relevance (see for example lines 154-159, 200-209, and much of the discussion section). A shorter, more focused presentation would have been more compelling.

Answer: We shortened the text as per reviewers' suggestions.

On balance, the research is very interesting. However, the modest yields, limited scope, and wordy presentation detract from the discovery. Given where related amide-forming reactions have been published, I am ambivalent about advocating for publication in Nat. Commun.

Answer: In this manuscript we are not selling our reaction as a 'one-size-fits-all' method for the synthesis of amides. The new reaction is highly innovative and unprecedented in terms of mechanism; but we also think it can be a highly useful reaction as it allows the synthesis of otherwise very difficult and expensively to access amides as elaborated in our manuscript. I would like to argue that for **every** established organic chemistry reaction plenty of examples can be found with modest yields, and limited scope – still they are published in high-ranking journals. '*Given where related amide-forming reactions have been published*' I completely disagree with this, and would like to ask for references for this opinion. Lastly, we mentioned in the outlook that other nucleophiles than water have the chance to extend this interesting new MCR by adding to the intermediate nitrilium species. Attempts to this are ongoing in my lab.

Reviewer #3 (Remarks to the Author):

This manuscript by Dr. Patil et al described "The Isocyanide SN2 Reaction". They discovered the weak nucleophilicity of the lone pair of electrons of the isocyanic carbon, which reacts with halides and water to form an SN2 reaction of amides through the high throughput

experimentation screening method, and optimized the reaction conditions. The substrate scope of the reaction and the post-modification of complex molecules to form amides were demonstrated using the optimized conditions in mild to good yield ; Meanwhile, the evidences for a related SN2 reaction mechanism is also partially provided. This is an important discovery in isocyanic chemistry and an effective complement to the method of synthesizing amides.

However, I have three major concerns to raise:

1) As the authors mentioned, there are many chiral compounds containing amide bonds in bioactive substances, therefore, due to the concern that the reaction conditions in the paper are relatively harsh, could the authors do a reaction to synthesize a peptide bond and observe whether its stereostructure is maintained?

Answer: Due to limitation in availability of starting material like chiral peptide containing halides we regret that we didn't carry out any examples as suggested by reviewer. However, in the similar context we tested the chiral halide containing starting material we reported this new example in the figure no. 3. We conclude that the inversion was observed in the molecules as we run the chiral separation of this product obtained by SFC and we found one isomer in major in proportion of (27:73). If this reaction was SN1 or radical type reaction then no inversion would be observed, instead a 1:1 mixture of both stereoisomers after the chiral separation of molecule. For the determination of the enantiomeric ratios, we employed chiral supercritical fluid chromatography coupled with mass spectrometry (SFC-MS) on the product. All experimental details are given in the SI.

2) In terms of mechanistic proof, could the authors use secondary halides as substrates (especially the carbon substituted by halogens are chiral carbons) to observe their Walden inversion phenomenon?

Answer: Yes, as suggested by reviewer we run the reaction. See the the previous answer, and yes, we observed the Walden inversion.

3) It is recommended to adjust some expressions in the discussion part, otherwise the experimental evidence cannot fully support the conclusion. For example, the authors mentioned "A chemoinformatic analysis of commercial building blocks demonstrates....." in page 11, If the authors have not done a verification experiment of chirality retention when the chiral compound forms an amide bond, it needs to be cautiously expressed.

Answer: With this particular statement we want show the differential possibilities of the

amide formation over the classical amidation reaction. We explained the cheminformatic analysis in detail in SI; we also added one chiral example of a chiral halide inside figure 3.

In addition there are some minor errors that need to be revised:

All changes were made accordingly.

1) In page 1, line 29, "the classic amid coupling"?

Answer: made changes accordingly.

2) In page 4, line 40, "by coupling of the"?

Answer: changed as " by the coupling of a"

3) In page 9, line 209, "of isocyanides (Fig.4)", is it Fig.5?

Answer: is it Fig.5, made changes accordingly.

4) In the supplementary part, S3, method development part, "We run he model reaction"?

Answer: Made changes accordingly "We run the model reaction"

5) In the supplementary part, S21, 4a, "3.43 (s, 1H)"?

Answer: Made changes accordingly: 3.43 (s, 2H) and 1.71 – 1.62 (m, 6H).

6) In the supplementary part, some spectra are not complete and clear. For example, S40, ¹HNMR of compound 3a, S41, ¹HNMR of compound 4a, et al. Please check all.

Answer: corrected

7) In the supplementary part, related solid compounds lack melting point data. In short, there are many writing and editing errors in this edition of the manuscript and SI, and the above may only be part of them, and the authors are expected to revise it carefully.

Answer: Melting points reported to the compounds as per their physical status after isolation.

This is an interesting work, however, the manuscript can only be considered for publication in Nature Communications after the points above being addressed thoroughly and an overall re-evaluation thereafter.

Answer: To all the reviewers a big 'thank you' for reviewing our manuscript and giving insightful and helpful comments.

Reviewers' Comments:

Reviewer #3:

Remarks to the Author:

I am satisfied with the authors' reply and this revised version.